# Improved Design of Slope-Shaped Hole-Blocking Layer and Electron-Blocking Layer in AlGaN-Based Near-Ultraviolet Laser Diodes

**DOI:** 10.3390/nano14070649

**Published:** 2024-04-08

**Authors:** Maolin Gao, Jing Yang, Wei Jia, Degang Zhao, Guangmei Zhai, Hailiang Dong, Bingshe Xu

**Affiliations:** 1State Key Laboratory of Integrated Optoelectronics, Institute of Semiconductor, Chinese Academy of Sciences, Beijing 100083, China; gml430493657@163.com (M.G.); dgzhao@red.semi.ac.cn (D.Z.); 2Key Laboratory of Interface Science and Engineering in Advanced Materials, Taiyuan University of Technology, Taiyuan 030002, China; jiawei@tyut.edu.cn (W.J.); zhaiguangmei@tyut.edu.cn (G.Z.); dhltyut@163.com (H.D.); 3Shanxi-Zheda Institute of Advanced Materials and Chemical Engineering, Taiyuan 030024, China; 4Center of Materials Science and Optoelectronics Engineering, University of Chinese Academy of Sciences, Beijing 100049, China

**Keywords:** UV laser, AlGaN, EBL, HBL, carrier

## Abstract

The injection and leakage of charge carriers have a significant impact on the optoelectronic performance of GaN-based lasers. In order to improve the limitation of the laser on charge carriers, a slope-shape hole-barrier layer (HBL) and electron-barrier layer (EBL) structure are proposed for near-UV (NUV) GaN-based lasers. We used Crosslight LASTIP for the simulation and theoretical analysis of the energy bands of HBL and EBL. Our simulations suggest that the energy bands of slope-shape HBL and EBL structures are modulated, which could effectively suppress carrier leakage, improve carrier injection efficiency, increase stimulated radiation recombination rate in quantum wells, reduce the threshold current, improve optical field distribution, and, ultimately, improve laser output power. Therefore, using slope-shape HBL and EBL structures can achieve the superior electrical and optical performance of lasers.

## 1. Introduction

Group Ⅲ nitrides have been extensively studied recently [1,2]. And AlGaN-based ultraviolet lasers have become a research hotspot in recent years due to their unique spectral range, small size, high efficiency, and long lifespan. They are widely used in air and water purification, ultraviolet sterilization and disinfection, automotive headlights, high-density light information storage, and other fields [3,4,5,6,7]. 

Compared with the typical InGaN blue laser structure, the quantum well energy in the active region of the AlGaN ultraviolet laser is shallower [8]. Moreover, the band offset in quantum wells and quantum barriers is very small [9], leading to a decrease in the carrier confinement ability of UV laser diodes (LD), severe electron and hole leakage, increased threshold current of UV-LD, and decreased optical output power [10].

In order to solve the above problems and optimize the limitation on charge carriers, many HBL and EBL structural designs related to the III-nitride LD have been proposed. For example, Zhang et al. proposed a structural design with an M-shaped hole barrier layer, which effectively reduces hole leakage in the n-type region [11]. Xu et al. proposed using inverted trapezoidal HBL to improve the light constraint factor of UVD-LD and reduce hole leakage [12]. EBL and HBL with gradient-based multi-quantum barrier (GMQB) structures can effectively alleviate severe polarization effects, thereby improving the injection ability of electrons and holes [13]. In addition, there are also designs for M-type HBL and W-type EBL, which can enable UVD-LED to achieve better optical output power and IQE [14]. The design of gradient rectangular superlattice (GRSL) EBL and gradient trapezoidal superlattice (GTSL) HBL can improve the radiation recombination rate of UVD-LD and reduce the threshold voltage [15]. In addition, there are also some structural designs related to the EBL and HBL of UVD-LD [16,17,18]. 

Due to the deep quantum well of GaN-based laser in the visible band, sometimes, it is difficult for holes to reach the first quantum well; generally, only EBL without HBL. Therefore, there is little literature on the design of HBL in UVA LDs. However, due to the shallow quantum well of the UVA-LD, holes are more likely to exceed the potential barrier of the quantum well to the n-side. Considering these issues, this paper proposes a slope shape HBL and EBL structure in UVA-LD. Three different shapes of HBL and EBL were compared with traditional UVA-LD, and the carrier concentration, effective barrier height, optical confinement factor, Auger recombination and stimulated radiation recombination in four LDs were studied using Crosslight LASTIP (2011 version) [19] simulation software. The results show that the optimized slope shape of HBL and EBL structures is more effective in limiting carrier leakage, increasing radiation recombination rate within the quantum well, reducing the threshold current, increasing the optical output power, and improving the optoelectronic performance of the LD compared to the traditional UVA-LD.

## 2. Laser Structure and Simulation Parameters

### 2.1. Laser Structure

The structures of the four proposed lasers diodes LD1-LD4 are shown in Figure 1. All four laser devices include 1 μm thick n-type GaN substrate *(n*-doping = 1.5 × 10^18^ cm^−3^); 0.6 μm thick n-type Al_0_._2_Ga_0_._8_N lower cladding layer (*n*-doping = 5 × 10^18^ cm^−3^); unintentionally doped Al_0_._08_Ga_0_._92_N lower waveguide layer with a thickness of 80 nm (*u*-doping = 5 × 10^16^ cm^−3^); a multi-quantum well consisting of three 10 nm thick Al_0_._07_Ga_0_._93_N quantum barrier layers and two 6 nm thick GaN quantum well layers sandwiched between the quantum barriers (*u*-doping = 5 × 10^16^ cm^−3^); 80 nm thick unintentionally doped Al_0_._08_Ga_0_._92_N upper waveguide layer (*u*-doping = 5 × 10^16^ cm^−3^); 0.5 μm thick p-type Al_0_._2_Ga_0_._8_N upper cladding layer (*p*-doping = 5 × 10^18^ cm^−3^); 20 nm thick p-type heavily doped GaN contact layer (*p*^+^-doping = 1 × 10^20^ cm^−3^). The top and bottom (black) parts are set as p-type electrodes and n-type electrodes, respectively. The EBL and HBL parameters of the four LDs are shown in Table 1. The laser emission wavelengths of 4 LDs are all around 360 nm.

### 2.2. Simulation Parameters

This paper uses LASTIP for simulation. Crosslight LASTIP (2011), a professional simulation software, is mainly dedicated to the 2D simulation of edge-emitting lasers. It includes optical gain models of quantum wells, quantum networks, Coulomb interactions, and kp non parabolic sub-bands that can be simulated. It can accurately simulate the interaction between multiple side modes of the laser beam. It can self-consistently solve Poisson’s equation, the current continuity equation, and drift diffusion model to analyze the optical and electrical characteristics of lasers [20,21,22]. The ridge width and cavity length of the four lasers are 3 μm and 600 μm, respectively. The working temperature is set to 300 K and the screening factor is set to 0.25. Only the Al components of HBL and EBL are changed; the remaining parameters in the laser structure remain unchanged. The formula for calculating the bandgap energy of Al_x_Ga_1-x_N ternary alloy is as follows [23]:(1)$$E_{g}({\mathrm{Al}}_{x}{\mathrm{Ga}}_{1-x}N)=3.45(1-\;x)+6.13x-\;1.3x(1-x),$$
where x is the Al component in AlGaN, and the direct bandgap energy of GaN is 3.45 eV.

Considering that the center Ga atom does not coincide with the positive and negative charge centers formed by its adjacent atoms in the simulation calculation process, a built-in electric field is generated, resulting in a spontaneous polarization effect. And the piezoelectric polarization effect caused by the lattice mismatch between AlGaN layers of different components, which leads to epitaxial stretching or compression, and intensifies or weakens the electric field formed by positive and negative charge centers. The physical performance of the laser is affected to a certain extent. The spontaneous polarization and piezoelectric polarization of ternary nitride alloys are obtained based on the Vegard interpolation method for binary nitrides. The expression [24] is (in C/m^2^ of the unit) as follows:(2)$${P^{\mathrm{sp}}}_{\mathrm{AlxGa}1-\mathrm{xN}}=-0.09x-\;0.034(1-\;x)+0.019x(1-x),$$
(3)$${P^{\mathrm{pz}}}_{\mathrm{AlN}}=-1.808\varepsilon +5.624\varepsilon^{2}\;,\varepsilon <0,$$
(4)$${P^{\mathrm{pz}}}_{\mathrm{AlN}}=-1.808\varepsilon -7.888\varepsilon^{2},\;\varepsilon >0,$$
(5)$${P^{\mathrm{pz}}}_{\mathrm{GaN}}=-0.918\varepsilon +9.541\varepsilon^{2},$$

ε as a function of base strain is as follows:(6)$$\varepsilon (x)=\frac{\left[a_{subs}-a(x)\right]}{a(x)},$$

a(x) represents the lattice constant of the unstrained alloy at component x and the lattice constant of the matrix.

The refractive index of Al_x_Ga_1-x_N changes with changes in laser wavelength, composition, and temperature. Under conditions such as a temperature of 300 K and a laser wavelength of 360 nm, the formula for calculating the refractive index of Al_x_Ga_1-x_N is as follows [25]:n(AlxGa1-xN) = [n(AlN)-n(GaN)]x + n(GaN), (7)
for the calculation of refractive index of graded Al component, the average refractive index of graded layer is currently used, and the formula is as follows [26]:(8)$$n_{a}=\frac{\left(n_{1}+n_{2}\right)}{2},$$
where n_1_ is the refractive index of the lowest component in the gradient component AlGaN, and n_2_ is the refractive index of the highest component in the gradient component AlGaN. In the formula, the refractive indices of GaN and AlN at 360 nm are 2.6538 and 2.2151, respectively. It is expected that the refractive index of the active region is higher than that of the adjacent cladding layer so as to form an effective optical waveguide effect, which will play a positive role in reducing the threshold current of the laser, beam divergence angle and the number of oscillation modes. The relative refractive index difference (Δn/n) should be 3–7% [27].

The Si impurity ionization energy of all materials mentioned in this paper is set to 20 meV, and the Mg impurity ionization energy of Al_x_Ga_1-x_N is set as follows:I = 170(meV) + 3x(meV),(9)
where x is the Al component in Al_x_Ga_1-x_N.

The absorption coefficients of n-type and p-type layers are set at 5 cm^−1^ and 50 cm^−1^, respectively. Considering the absorption of free carriers, the absorption coefficients of p-type layers with different doping concentrations are also different, which are set according to the following formula [28]:(10)$$\alpha_{i}=\frac{doping\;concentration({cm}^{-3})}{10^{19}({cm}^{-3})}\times 25({\mathrm{cm}}^{-1})$$

## 3. Results and Discussion

Figure 2a shows the modelled P-I curves of four lasers at an injection current of 120 mA. As shown in Figure 2a, when the injection current increases to 39.45 mA, the output power of LD4 rises sharply first, and then the output power of the other three LDs also rises sharply. Compared with the reference traditional structure LD1, the threshold current of the improved three LDs is lower than that of LD1, and the output power is higher than that of LD1, which indicates that the slope shape HBL and EBL structure has improved the electrical performance of the laser. Figure 2b shows the threshold current and output power of the four lasers when the injection current is 120 mA. The threshold current of traditional LD1 is 46.66 mA, which is also the highest among the four lasers. The threshold current of LD2 is 43.20 mA, which is slightly lower than that of LD1. The threshold current of LD3 is 42.00 mA, which is lower than that of LD2. The threshold current of LD4 is the lowest at 39.45 mA, which is about 15.5% lower than that of LD1. By comparing the output power of the four lasers, it can be found that the output power of LD1 is 147.8 mW, which is the lowest among the four lasers, and the output power of the other three lasers is higher than that of LD1. The output power of LD2 is 154.7 mW; the output power of LD3 is 157.3 mW; the output power of LD4 is 171.5 mW, which is about 16% higher than that of LD1. The optical output power of the improved three LD is improved.

The threshold current of the laser largely depends on the gain and loss of light, which can be obtained by the following formula [29]:(11)$$J_{\mathrm{th}}=J_{t}+\frac{1}{\Gamma A}[\alpha_{ί}+\frac{1}{L}\ln\frac{1}{R}],$$
where Γ is the optical confinement factor, which represents the loss caused by the optical field expanding out of the active region; R is the reflectivity of the cavity surface, which is the product of the reflection R_1_ and R_2_ from the two end faces of the laser. We set the reflectivity of the two end faces to 90% and 10%, respectively, so R is a fixed parameter; α_ί_ is the internal loss of the medium; L is the length of the resonant cavity, which is a fixed value; J_t_ is the intercept of the gain curve in the current density coordinate. Through the LASTIP simulation, the gain curve, Γ and α_ί_ can be obtained directly. The gain curves of the four LDs were extracted, and it was found that the gain near the 360 nm wavelength basically did not change, while J_t_ was exactly extracted from the point where gain curve intercepts the axis of x = 0, which was basically the same for 4 LDs. Removing these fixed parameters, J_th_ has a close relationship with Γ and α_ί_, and the four LDs and α_ί_ are shown in Figure 3a. The sum of mirror loss and internal loss is the total loss of LD. In the calculation process, the mirror loss is a fixed value, so the value of internal loss also simulates the total loss of LD. The α_ί_ of LD2 is 7.284 (1/cm), LD3 is 7.259 (1/cm). They are lower than 7.349 (1/cm) of LD1. And the α_ί_ of LD4 is 6.77 (1/cm), which is much lower than the other three ones. Since the activation energy of Mg in p-type doped AlGaN will change with the change of Al composition, the higher the Al composition, the greater the activation energy of Mg [30]. Therefore, the composition of EBL in LD1 is 35%, while the average composition of EBL of LD4 is lower than 35%. Therefore, the Mg activation energy of LD4 is smaller than that of LD1. So, the ionization rate of Mg is higher and the absorption loss of LD4 is lower than that of LD1.

And we can clearly see that the Γ value of LD1 to LD4 also changes, which is related to the thickness of the active region and the optical field distribution [31]. The thickness of the active region of the four lasers is the same. The optical field distribution is shown in Figure 3b. Since LD1 has no HBL, the MQWs area is different from the other three LDs. The blue bar in the figure is the MQWs area of LD2, LD3 and LD4, and the green bar is the MQWs area of LD1. The optical field distributions of LD2 and LD3 are similar, while the optical field distribution of LD3 shrinks inward closer to the MQWs region. The optical fields of LD2 and LD3 are suddenly concave in the HBL region (indicated by the blue arrow in the figure), which may be the reason why the optical confinement factors of LD2 and LD3 are lower than those of LD1. The optical field distribution of LD4 is different from the other three. In the p-side region, the optical field moves more to the n-side and is far away from the p-side, which absorbs light. Compared with LD2, the maximum moving distance is 183.7 nm, but this also makes the region with optical intensity higher than 99% deviate from the MQWs region. These two factors have not improved the optical confinement factor of LD4 compared with LD1, and the optical confinement factors of LD4 and LD1 are basically the same. Therefore, the Γ value of LD4 has little change compared with LD1 and α_ί_ is the smallest of the four LD, which reduces the threshold current of LD4.

In addition, the problem of carrier leakage in four LDs was also studied. Figure 4 shows the distribution of carrier concentration in the four LDs, and the inset in the figure shows the distribution of hole concentration and electron concentration in the first quantum well. The distribution of hole concentration is shown in Figure 4a. LD1 does not have HBL, so HBL (purple area in the figure) in the lower area label is the lower waveguide layer (LWG) of LD1. The parameters of LD3 and LD4 in each layer of the n-side area are the same, so the hole concentration distribution of the two overlaps. It can be clearly seen from the figure that the hole concentrations of LD2, LD3, and LD4 in LWG are lower than those of traditional LD1, which also indicates that these three structures can effectively limit hole leakage into the n-side LWG compared to traditional LD1. The electron concentration distribution is shown in Figure 4b, and the parameters of each layer in the p-side region are the same for LD1, LD2 and LD3, so their electron concentration distribution curves overlap. It can be clearly seen from the figure that the peak at the interface between the upper waveguide layer (UWG) and EBL of LD4 is the lowest, which is attributed to the polarization effect, indicating that LD4 has the least electron accumulation at this point. And, it can be seen that the electron concentration of LD4 in the upper cladding layer (UCL) is also the lowest. This also means that LD4 leaks the fewest electrons to the UCL, and LD4 can effectively prevent electron leakage. Electronic leakage can lead to the recombination of leaked electrons with p-type region holes, resulting in carrier loss and an increase in threshold current. In addition, the illustration in Figure 4 shows the carrier concentration in the first quantum well, and it can be observed that the hole concentration and electron concentration are both lower in LD4 than in LD1. This also indicates that the recombination rate of holes and electrons in LD4 quantum wells is higher than the other three.

The holes and electrons in semiconductor lasers are transported through drift motion under the influence of an electric field, and diffusion motion under concentration gradient, resulting in hole current and electron current, as shown in Figure 5. The hole current density of the four LDs is shown in Figure 5a; the hole leakage current of LD1 is higher than that of the other three, which indicates that adding a layer of HBL to the structure of DUA-LD can limit the leakage of holes. And the hole-injection current of LD4 is significantly higher than that of LD1. An increase in the p-side hole-injection current means that more holes are transported from the p-side to the active region, improving the injection efficiency of holes and increasing the probability of radiation recombination between holes and electrons in the active region. Figure 5b shows the electron current density. Before the electron current passes through the active region, the electron current density of LD4 is significantly higher than that of LD1, which also means that LD4 injects more electrons into the active region than LD1, thereby increasing the probability of radiation recombination. When the electron current reaches the p-side layers, the electron leakage current of LD4 is lower than that of LD1 (the black curve of LD1 coincides with the blue curve of LD3), especially in the EBL, where the leakage current of LD1 shows a sudden increase trend. When a forward bias is applied to the laser diode, electrons are injected from the n-type region into the active region. Due to the high mobility and small effective mass of electrons, some electrons will gain sufficient energy to overflow from the active region and flow towards the p-type region. The overflow electrons recombine with holes in the p-type region, forming an electron leakage current [32]. The electronic leakage current does not participate in stimulated radiation, which would reduce the internal quantum efficiency. This indicates that the structure of LD4 can effectively prevent carrier leakage and promote carrier injection.

In order to understand the difference between carrier leakage and injection in four LDs, the energy band diagrams of the four LDs were plotted at an injection current of 120 mA, as shown in Figure 6. Due to lattice mismatch, piezoelectric and spontaneous polarization fields, the energy band will bend and deviate from the ideal state [33]. The p-side layers of LD1, LD2 and LD3 are the same, so their energy bands in the p-side region are similar. And the n-side layers of LD3 and LD4 are the same, so the energy bands of LD3 and LD4 in the n-side region are similar. The electrons in the conduction band are more inclined to move to the region with lower energy levels, while the conduction band of the EBL rises upward, which can block the transport of electrons to the UCL. Meanwhile, in the valence band, holes tend to move towards higher energy levels. The valence band of the HBL is concave, which can suppress hole leakage into the LCL. To analyze the band differences between LD more clearly, we will enlarge the band diagrams of the HBL and EBL of some LDs.

Firstly, analyze the energy band difference between LD1 and LD4 from the conduction band. The effective barrier height of electrons is defined as the gap between the highest conduction band energy level in EBL and HBL and the Fermi level in the left layer, marked with blue arrows and numbers in the figure. The enlarged image of some areas in the conduction band is shown in Figure 7. LD1 does not have HBL, so the HBL band of LD2 is compared with that of LD4. The HBL of LD2 has a prominent protrusion (circled in red in the figure), which significantly increases the effective electron barrier height of the EBL to 133 meV. An increase in the effective barrier height of electrons means an increase in the blocking effect on electrons and a decrease in the concentration of electrons injected into the active region. LD4 can effectively remove this protrusion in the HBL, thereby reducing the effective potential barrier height of electrons to 72 meV, a decrease of 45.87%; thus, increasing the electron injection efficiency. Analyzing the conduction band of the EBL, the energy band at the interface between the UWG and the EBL of LD1 is concave downwards (circled in green in the picture), which leads to electron accumulation at the interface, electron loss, and an increase in threshold current. The LD4 with a slope shape structure smoothes out this concave channel and reduces electron stacking. The effective electron barrier heights of LD1 and LD4 are 192 meV and 228 meV, respectively. The effective barrier height of LD4’s electrons has increased compared to LD1, indicating that LD4 can effectively block electron leakage.

Next, we will analyze the valence bands of the HBL of LD2 and LD4. The effective barrier height of holes is defined as the gap between the highest conduction band energy level in EBL and HBL and the Fermi level in the right layer, marked with blue arrows and numbers in the figure. The enlarged image of the valence band is shown in Figure 8, and the effective barrier height of the holes is marked with blue numbers in the figure. The slope shape component of LD4 causes a significant change in the energy band compared to LD2, transitioning from the smooth energy band of LD2 to the stepped shape, and resulting in a change in the effective potential barrier height of the holes. For the HBL, the effective hole barrier heights of LD4 are 350 meV, which are higher than the 342.6 meV of LD2. This means that it is more difficult for holes to cross the HBL of LD4 and reach the LCL, as LD4 can effectively block hole leakage. For the EBL, the effective hole barrier heights of LD4 are 166 meV, slightly lower than the 168 meV of LD1. From this, it can be inferred that there is a slight difference in hole injection rates between the two. The slope shape of LD4 can improve hole injection, which also explains why the hole current density of LD4 in Figure 5 is higher than that of LD1, increasing the concentration of holes flowing into the quantum well region.

We further analyze the Auger recombination rate and stimulated radiation rate of four LD and two quantum wells, as shown in Figure 9. Auger recombination is a non-radiative recombination that is the inverse process of collision ionization, which greatly reduces the lifetime of minority carriers [34], thereby increasing the threshold current of the laser and reducing the output power. In Figure 9a, it can be visually observed that the Auger recombination rate of the second quantum well is slightly higher than that of the first, and, in both quantum wells, the peak Auger recombination rate of LD4 is lower than that of LD1 (the black number in Figure 9a indicates the peak of the Auger recombination rate); the peak Auger recombination rate of LD3 is highest in the second quantum well. LASTIP can only calculate Auger recombination in non-radiative recombination, so Auger recombination replaces nonradiative recombination. This also means that the nonradiative recombination of LD4 is lower than that of LD1, and the carrier lifetime is higher than that of LD1, ultimately resulting in a lower threshold current of LD4 than LD1. The stimulated radiation is a necessary condition for generating lasers and also the main optical source of lasers. As shown in Figure 9b, the stimulated radiation recombination rate of LD4 in both quantum wells is higher than that of LD1, especially in the first quantum well; LD4 is more than twice as high as LD1, and it is the highest among the four LDs. This phenomenon further explains the reason why there are fewer electrons and holes in the first quantum well of LD4 in Figure 4. This also indicates that the stimulated radiation occurring within LD4 is much higher than that of LD1, and more photons are emitted due to the stimulation of a certain amount of electrical energy compared to LD1. The result on the P-I curve is that the optical output power of LD4 is much higher than that of LD1. Overall, LD4 has the lowest nonradiative recombination rate, resulting in a lower threshold current, while the highest stimulated radiation rate brings higher optical output power.

## 4. Conclusions

In summary, we systematically studied the effects of different HBL and EBL on the optoelectronic properties of GaN-based UVA-LDs. Research has found that, among the four LDs with the same MQWs, the threshold current of the three LDs with HBL is lower than that of the traditional LD without HBL, and the output power is higher than that of the traditional LD. The LD with a fixed component HBL has a protrusion in the conduction band at the interface between HBL and the first quantum barrier, which affects the injection of electrons. At the interface between UWG and EBL, there is a concave channel in the conduction band, which generates electron stacking. However, changing only the fixed component of HBL to the slope shape component of LD3 can optimize the energy band of HBL; but this increases the Auger recombination rate and decreases the stimulated radiation recombination rate. So, the LD of HBL and EBL with slope shape components are designed, which have superior effective barrier heights compared to the other three LD energy bands. They can reduce carrier leakage and improve carrier injection efficiency. The stimulated radiation recombination rate in their quantum wells is much higher than that of traditional LD. And, due to its unique composition changes, the refractive index also changes, causing its optical field to move away from the p-side, thereby achieving superior optoelectronic performance.

## Figures and Tables

**Figure 1 nanomaterials-14-00649-f001:**
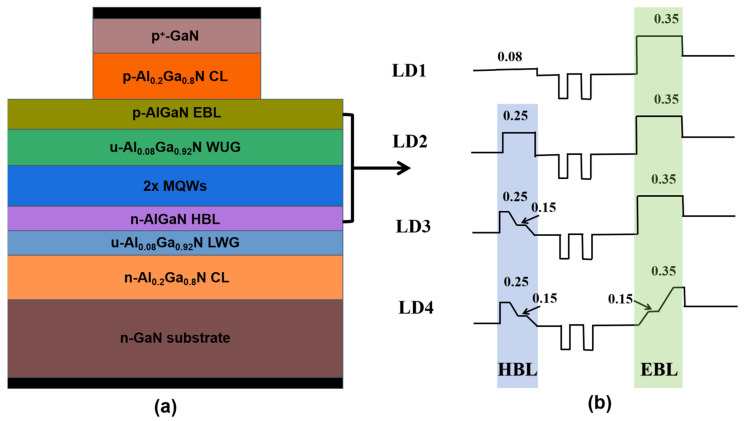
Schematic diagram of four types of lasers (**a**) and schematic diagram of Al component changes in HBL and EBL (**b**).

**Figure 2 nanomaterials-14-00649-f002:**
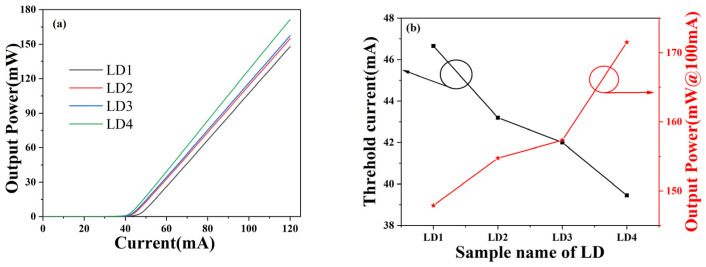
P-I curves (**a**) and output power and threshold current (**b**) at 120 mA of four lasers.

**Figure 3 nanomaterials-14-00649-f003:**
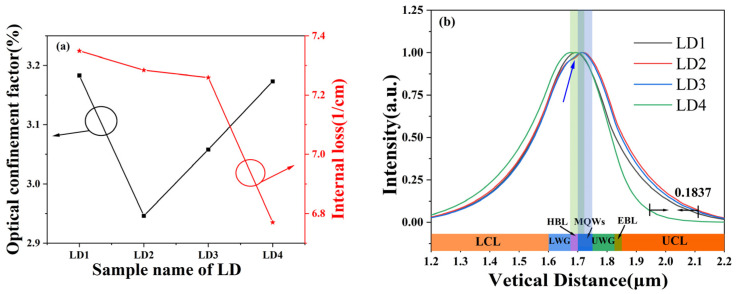
Optical confinement factor Γ and internal loss α_ί_ (**a**) and optical field distribution (**b**) (the blue bar in the (**b**) is the MQWs area of LD2, LD3 and LD4, and the green bar is the MQWs area of LD1).

**Figure 4 nanomaterials-14-00649-f004:**
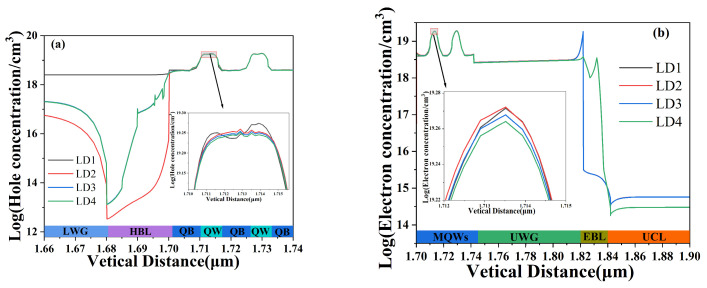
Hole concentration distribution (**a**) and electron concentration distribution (**b**) of four LDs under the injection current of 120 mA. The inset in the figure shows the distribution of hole concentration and electron concentration in the first quantum well.

**Figure 5 nanomaterials-14-00649-f005:**
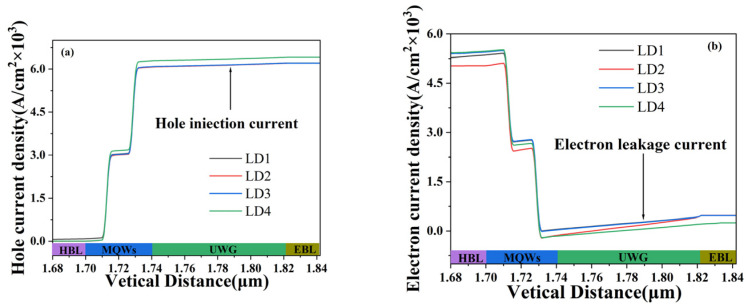
Hole current density (**a**) and electron current density (**b**) of four LDs.

**Figure 6 nanomaterials-14-00649-f006:**
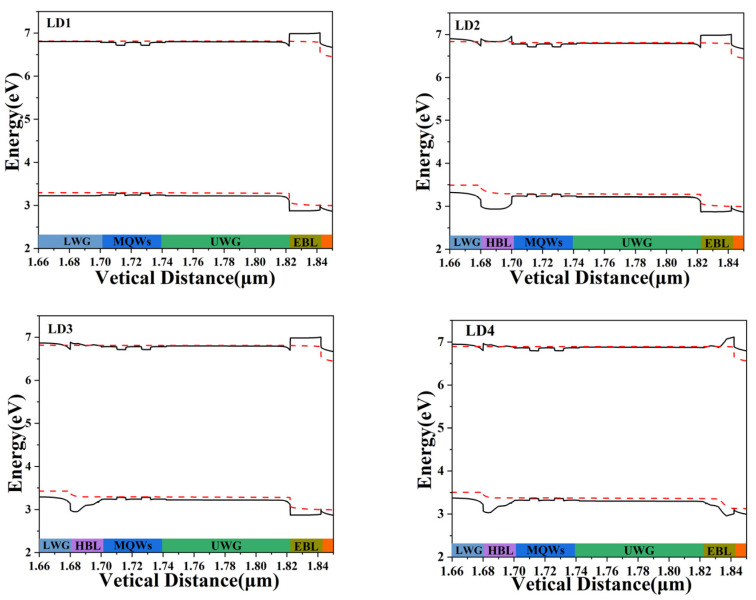
Energy band diagrams of four LDs.

**Figure 7 nanomaterials-14-00649-f007:**
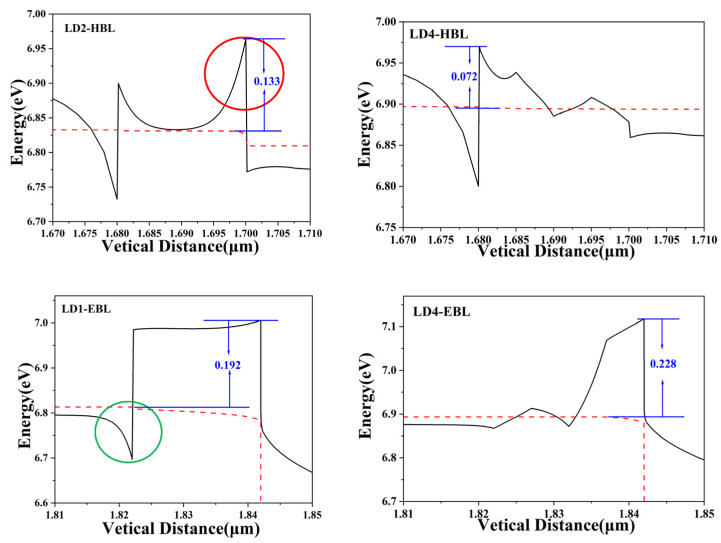
Enlarged view of EBL and HBL in partial LD conduction band.

**Figure 8 nanomaterials-14-00649-f008:**
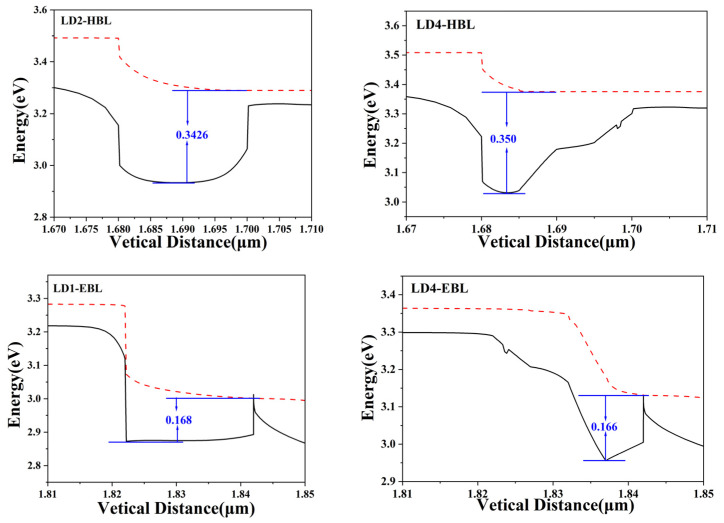
Enlarged images of EBL and HBL in some valence bands.

**Figure 9 nanomaterials-14-00649-f009:**
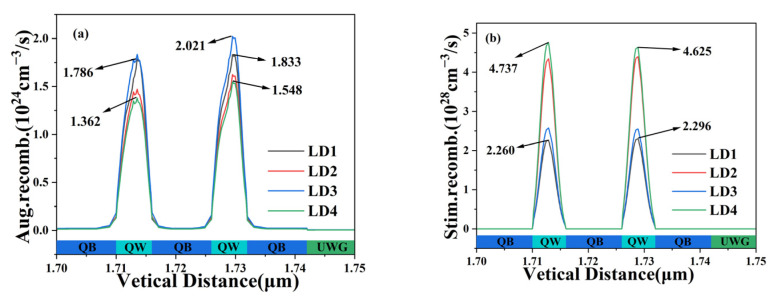
Auger recombination rates (**a**) and stimulated radiation recombination rates (**b**) of four LDs.

**Table 1 nanomaterials-14-00649-t001:** Parameters of EBL and HBL layers for four LDs.

The Names of the Laser	HBL of Al Composition (%)	HBL of Layer Thickness (nm)	EBL of Al Composition (%)	EBL of Layer Thickness (nm)
LD1	0	0	35	20
LD2	25	20	35	20
LD3	25	5	35	20
	25 → 15	5		
	15	5		
	15 → 7	5		
LD4	25	5	7 → 15	5
	25 → 15	5	15	5
	15	5	15 → 35	5
	15 → 7	5	35	5

## Data Availability

The data presented in this study are available on request from the corresponding author upon reasonable request.

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
