# Peer review of "Improved Design of Slope-Shaped Hole-Blocking Layer and Electron-Blocking Layer in AlGaN-Based Near-Ultraviolet Laser Diodes"

_nanomaterials, 2024, doi:10.3390/nano14070649_

Round 1

Reviewer 1 Report

Comments and Suggestions for Authors

This manuscript reports on the design of slope shape hole barrier layer (HBL) and electron barrier layer in AlGaN based near-ultraviolet lasers improves the optoelectronic performance of laser diodes. The idea is to achieve superior electrical and optical performance of the AlGaN structures. Results obtained are valuable and could be directly used for the advancement pf the application of group III nitride materials for the purposes of LDs. Sampling is well chosen and the LASTIP simulations are very well followed and analyzed. Consequently, this work is original methodologically but also directly usable for further studies. I see as a positive point also the clarity of discussion around the LASTIP simulations.

The work raises only some minor textual concerns related to contextualization of the present results. These points amount to a minor revision before the acceptance of manuscript for publication:

1: Title is too long, and it should not include acronyms.

2: Abstract: Abstract should shortly but explicitly refer to the simulation technique used, namely, LASTIP.

3: Have the authors considered existing theoretical/experimental results as per the wide variety of group III nitrides and corresponding semiconductor structures especially those underlining their electronic and optical properties, design and their corresponding applications?

4: In relation to the above existing, there is literature dedicated to the properties of group III nitrides and corresponding semiconductor structures using ab initio methods as guidance for improving their properties, namely, ACS Nanosci. Au 2023, 3, 1, 84–93, and CrystEngComm 25 (2023) 5810-5817. Such recent works should be reflected in the introduction.

Comments on the Quality of English Language

The manuscript still needs a comprehensible stylistic and grammatical revision.

Author Response

Dear Reviewer

Thank you very much for your email on 29 Mar 2024 about our manuscript (nanomaterials-2952363). Based on your email, we carefully read the your comments. We appreciate your comments which have helped us in improving the quality of our manuscript. In the enclosed “Response to the Reviewer’s Comments”, we respond these comments carefully.

Sincerely yours,

Maolin Gao

Reviewer 2 Report

Comments and Suggestions for Authors

Review of manuscript nanomaterials-2952363 by Gao et al.
1. The title implies that the recommended design has actually improved optoelectronic device performance of AlGaN. This has not been verified because the manuscript only deals with simulations, which should be clearly stated. So, the words ‘improves the’ should be replaced by ‘may improve’. Also in the title, ‘slope shape’  should be replaced by ‘slope-shaped’, ‘HBL and EBL’ by ‘hole and electron blocking layers’ and ‘LD’ by ‘laser diodes’ to remove acronyms.
2. Replace the imperative ‘Use’ by the words ‘We used’ in line 17 and include a reference to the software. In line 57 the authors mention a different name for the software (‘Lastip’?). Please clarify and support with references.
3. Replace ‘Research has found’ in line 18 by ‘Our simulations suggest’, and replace ‘can’ in line 22 by ‘may’ or ‘could’.
4. Correct statement in line 27/28: the operation threshold voltage for typical AlGaN UV lasers is not really low at all.
5. Define acronym LD when first used.
6. Replace semi-colons in lines 41, 43 and 45 by full stops (periods).
7. The statement in lines 50/51 that ‘the design of UVA-LD adjacent to the visible light band is relatively limited in terms of HBL ’ is unclear - do the authors mean that HBLs are limited to UVA-LDs instead?
8. Do not capitalise the adjective ’stimulated’ in lines 56 and line 334, 347, 348, 352, 357 and 361.
9. The statement in line 65 ‘ware grown on a C-plane’ is both grammatically and technically incorrect because for simulations nothing is ‘grown’. The letter ‘c’ should be italics and not capital. The enumeration of layers needs either a list or proper grammar. Essentially, structures LD3 and LD4 contain graded AlGaN layers wherein the aluminium content is ramped up/down by +20%/–10% over a distance of only 5nm. At what temperature could such a gradient be realized in practice given the known segregation length of about 1.4nm of Al in AlGaN at 1400K? Figure 1 seems to contain double rather than single gradients in each graded AlGaN HBL/EBL layer?
10. Equation (1) gives the bandgap of GaN (x=0) as 3.45 (presumably in eV), not 3.39eV as stated. Is the difference relevant?
11. What are the units of P in equations (2-5)?
12. Insert ‘modelled’ before ‘P-I curves’ in line 153.
13. Quoting output powers in section 3 of modelled laser structures in the 100mW range to 2 or 3 digits after the comma makes no sense.
14. Please check the x-axis labels of figures 2a (partially hidden), 3b-9 (spelling of ‘vertical’) and the y-axis labels of 3b (units?).
15. What are ‘LWG’ in lines 223, 225, ‘UWG’ in line 228, ‘UCL’ in lines 231, 232?
16. What is meant by ‘LASTIP software believes that the other types of Nonradiative recombination are the same’ in line 343?
17. Do not capitalise the adjective ’nonradiative’ in lines 343, 344, 356.

Main issues:
A. Clarify where and how do the presented simulations and their results different from those in references [9-17], i.e. what is the novelty?
B. How physical are the many approximations used, e.g. the reflectivity R in equation (11) is treated as a constant, which is only correct for vertical incidence, without any polarisation and without any dispersion (-> Fresnel’s formulas!). There are many other simplifications which need to be stated. 

Comments on the Quality of English Language

mostly OK, see points 2.a, 6. ,8., 9.a, 17.

Author Response

(The authors gave the same response as above.)
